# A_2B_ Adenosine Receptors: When Outsiders May Become an Attractive Target to Treat Brain Ischemia or Demyelination

**DOI:** 10.3390/ijms21249697

**Published:** 2020-12-18

**Authors:** Elisabetta Coppi, Ilaria Dettori, Federica Cherchi, Irene Bulli, Martina Venturini, Daniele Lana, Maria Grazia Giovannini, Felicita Pedata, Anna Maria Pugliese

**Affiliations:** 1Department of Neuroscience, Psychology, Drug Research and Child Health (NEUROFARBA), Section of Pharmacology and Toxicology, University of Florence, 50139 Florence, Italy; ilaria.dettori@unifi.it (I.D.); federica.cherchi@unifi.it (F.C.); irene.bulli@unifi.it (I.B.); martina.venturini@unifi.it (M.V.); felicita.pedata@unifi.it (F.P.); annamaria.pugliese@unifi.it (A.M.P.); 2Department of Health Sciences, Section of Clinical Pharmacology and Oncology, University of Florence, 50139 Florence, Italy; daniele.lana@unifi.it (D.L.); mariagrazia.giovannini@unifi.it (M.G.G.)

**Keywords:** cerebral ischemia, oxygen-glucose deprivation, neuroinflammation, A_2B_ receptors, oligodendrocyte differentiation, demyelination, adenosine

## Abstract

Adenosine is a signaling molecule, which, by activating its receptors, acts as an important player after cerebral ischemia. Here, we review data in the literature describing A_2B_R-mediated effects in models of cerebral ischemia obtained in vivo by the occlusion of the middle cerebral artery (MCAo) or in vitro by oxygen-glucose deprivation (OGD) in hippocampal slices. Adenosine plays an apparently contradictory role in this receptor subtype depending on whether it is activated on neuro-glial cells or peripheral blood vessels and/or inflammatory cells after ischemia. Indeed, A_2B_Rs participate in the early glutamate-mediated excitotoxicity responsible for neuronal and synaptic loss in the CA1 hippocampus. On the contrary, later after ischemia, the same receptors have a protective role in tissue damage and functional impairments, reducing inflammatory cell infiltration and neuroinflammation by central and/or peripheral mechanisms. Of note, demyelination following brain ischemia, or autoimmune neuroinflammatory reactions, are also profoundly affected by A_2B_Rs since they are expressed by oligodendroglia where their activation inhibits cell maturation and expression of myelin-related proteins. In conclusion, data in the literature indicate the A_2B_Rs as putative therapeutic targets for the still unmet treatment of stroke or demyelinating diseases.

## 1. Introduction

### Adenosine as a Signaling Molecule

Adenosine is a naturally occurring nucleoside belonging to one of the oldest signaling pathways, the purinergic system, involved in a variety of physiological and pathological processes [1]. In the CNS, adenosine is formed intracellularly from adenosine monophosphate (AMP) degradation, particularly under high energy demand, or extracellularly by the metabolism of released nucleotides operated by membrane-bound ecto-enzymes like CD39 and CD73 [2]. A vesicular mechanism of adenosine release in an excitation–secretion manner has also been postulated [3,4]. Extracellular adenosine is removed by enzymes devoted to its degradation, such as adenosine deaminase (ADA) or adenosine kinase (AK) [5], or taken up by the equilibrative nucleoside transporter (ENT) isoforms ENT1 and ENT2 [6]. Enhanced extracellular concentrations of adenosine can be considered as a general harm signal, contributing to the recruitment of damage-associated molecular effectors [2].

Adenosine acts through the activation of four different purinergic P1 receptors: A_1_, A_2A_, A_2B_, and A_3_ adenosine receptors (A_1_Rs, A_2A_Rs, A_2B_Rs, and A_3_Rs, respectively), all belonging to the G-protein coupled, metabotropic receptor family [7]. Adenosine signaling through P1 receptors has long been a target for drug development, with adenosine itself or its derivatives being used clinically since the 1940s. Methylxanthines such as caffeine and theophylline have profound biological effects as antagonists at adenosine receptors [5]. Moreover, drugs such as dipyridamole and methotrexate act by enhancing the activation of adenosine receptors [8,9].

The most widely recognized adenosine signaling is through the activation of A_1_Rs, which inhibits adenylyl cyclase (AC) through G_i/o_ protein activation [10]. A_1_Rs are widely distributed in most species, and mediate diverse biological effects. They are dominant in the central nervous system (CNS), with high levels being reported in the cerebral cortex, hippocampus, cerebellum, thalamus, brainstem, and spinal cord, where they inhibit neurotransmission by different mechanisms: (1) they decrease glutamate release by inhibiting presynaptic voltage-gated Ca^2+^ channels (VGCC) [11,12] and (2) they stabilize neuronal membrane potential by increasing K^+^ and Cl^-^ conductances at the postsynaptic site [13]. Consistently, A_1_R stimulation has been implicated in sedative, anticonvulsant, anxiolytic, and locomotor depressant effects in the CNS, whereas, at cardiovascular levels, they are potent bradycardic agents [14].

The A_2A_R subtype is known to stimulate AC [10] being coupled to G_s_ proteins [7]. This receptor subtype was isolated from a human hippocampal cDNA library [15] and, in the brain, it is also highly expressed in the striatum/caudate-putamen nuclei [16], whereas, in the periphery, high expression has been observed in immune tissues [17]. At central level, the functional effect of A_2A_R activation is at variance from A_1_Rs, as they are reported to enhance glutamate release by facilitating Ca^2+^ entry through presynaptic VGCC and inhibiting its uptake [18]. Moreover, A_2A_R inhibits voltage-dependent K^+^ channels, thus promoting cell excitability and neurotransmitter release [19]. Concerning peripheral functions of A_2A_Rs, it is worth noting that adenosine, thanks to its actions on this receptor subtype, is one of the most powerful endogenous anti-inflammatory agents [7]. Indeed, A_2A_Rs are highly expressed in inflammatory cells including lymphocytes, granulocytes, and monocytes/macrophages, where their activation reduces pro-inflammatory cytokine production, i.e., tumor necrosis factor-alpha (TNFα), interleukin-1β (IL-1 β), and IL-6 [20] and enhances the release of anti-inflammatory mediators, such as IL-10 [21].

The relatively new A_3_R subtype, cloned in 1993 [22], is coupled to G_i/o_ proteins and inhibits AC but, under particular conditions or in different cell types, activation of G_q/11_ by A_3_R agonists has also been reported [7], with consequent increase in intracellular [IP_3_] and [Ca^2+^]. The A_3_R shows large interspecies differences, with only 74% sequence homology between rat and human [23]. Its expression is not uniform throughout the body—low levels are found in the brain and spinal cord, whereas a predominance of this receptor subtype is described in peculiar regions at the periphery, i.e., in the testis, lung, kidneys, placenta, heart, brain, spleen, and liver [24]. Interestingly, most of the cell types of the immune system express functional A_3_Rs on their surface [25] and its activation is one of the most powerful stimuli for mast cell degranulation [26].

## 2. A_2B_ Adenosine Receptors (A_2B_Rs)

This adenosine receptor subtype is somewhat the most enigmatic and less studied among the four P1 receptors. Although it was cloned in 1995 [27], a pharmacological and physiological characterization of A_2B_Rs has long been precluded by the lack of suitable ligands able to discriminate among the other adenosine receptor subtypes [28].

The distribution of A_2B_Rs in the CNS on neurons and glia is scarce but widespread, whereas in the periphery, abundant expression of A_2B_Rs is observed in the bronchial epithelium, smooth muscles, mast cells, monocytes, and digestive tracts such as ileum and colon [29]. Functional A_2B_Rs have been also recognized in vascular beds, fibroblasts, and hematopoietic and neurosecretory cells [30]. The activation of A_2B_Rs stimulates G_s_ and, in some cases, G_q/11_ proteins, thus enhancing intracellular [cAMP] or [IP_3_], respectively [7]. As mentioned above for the cognate A_2A_R subtype, in addition to brain cells and endothelial cells, A_2B_Rs are present on hematic cells, such as lymphocytes and neutrophils, with the highest expression levels on macrophages [31,32]. Here, A_2B_ receptors in most cases are coexpressed with A_2A_Rs and their activation exerts anti-inflammatory effects, inhibiting vascular adhesion [32] and migration of inflammatory cells [33]. Thus, attenuation of hypoxia-associated increases of tissue neutrophils due to infiltration in different tissues including the brain, may largely depend on blood cell A_2B_R signaling [34].

Differently from the high affinity A_1_Rs, A_2A_Rs and A_3_Rs, which are activated by physiological levels of extracellular adenosine (low nM and high nM, respectively [35]), the A_2B_R needs much higher adenosine concentrations (in the µM range) reached only in conditions of tissue damage or injury. Such a low affinity of A_2B_Rs for the endogenous agonist implies that they represent a good therapeutic target, since they are activated only by high adenosine efflux reached under pathological conditions or injury, when a massive release of adenosine occurs [35,36] or that they can be driven to function by selective agonists

## 3. A_2B_ Adenosine Receptors (A_2B_Rs) in the Hippocampus

Similarly to the A_2A_R subtype, A_2B_R activation within the CNS is reported to increase glutamate release [37,38]. However, a distinct mechanism has been described. Indeed, Cunha and co-workers demonstrated that the A_2B_R selective agonist BAY60-6583 attenuates the predominant A_1_R-mediated inhibitory control of synaptic transmission in the CA1 hippocampus [37]. These data are consistent with the relatively abundant expression of A_2B_Rs in hippocampal synaptosome preparations reported by the authors [37]. The facilitatory effect of A_2B_Rs on glutamatergic neurotransmission was assessed in acute hippocampal slices using the electrophysiological protocol of paired pulse facilitation (PPF), which is known to modulate short-term synaptic plasticity. Our group of research recently confirmed that A_2B_Rs decreases PPF, thus enhancing glutamate release, in an A_1_R-dependent manner. Indeed, the effect of BAY60-6583 was prevented not only by the A_2B_R antagonists MRS1754 and PSB-603, but also by the A_1_R blocker DPCPX [38]. Furthermore, we extended results to a newly synthetized BAY60-6583 analogue, the A_2B_R-selective agonist P453 recently described [39], which proved to have higher affinity than BAY60-6583 [38].

The fact that neither A_2B_R agonists [37,38] nor antagonists [40] affect basal hippocampal glutamatergic transmission suggests that the role of A_2B_Rs might be confined to conditions of synaptic plasticity. Nevertheless, the effect of A_2B_R activation on PPF should not be underestimated as it could be crucial to cognitive performances and mechanisms related to memory and learning [41]. Indeed, it was demonstrated that compounds able to facilitate a PPF ratio, as A_2B_R agonists do, may improve cognitive processes and memory performances in behavioral tests in rodents [42].

## 4. A_2B_Rs and Oligodendrogliogenesis

We recently and originally demonstrated that A_2B_Rs are crucially involved in oligodendrocyte progenitor cell (OPC) maturation. We found that the selective A_2B_R agonists BAY60-6582 (10 μM) and P453 (500 nM) inhibited the differentiation of purified primary OPC cultures, as demonstrated by the reduced expression of myelin-related proteins such as myelin basic protein (MBP) or myelin associated glycoprotein (MAG). We also demonstrated that A_2B_R activation reversibly inhibits tetraethylammonium- (TEA-) sensitive, sustained I_K_, and 4-amynopyridine- (4-AP) sensitive, transient I_A_, conductances [43]. As I_K_ are known to be necessary to OPC maturation [44], this could be one of the mechanisms by which A_2B_Rs inhibit myelin production. These results are similar to what was observed in cultured OPCs exposed to the A_2A_R agonist CGS21680, as demonstrated by us in a previous work [45,46,47]. At variance, the activation of a G_i_-coupled, P2Y-like receptor GPR17, recently deorphanized [48,49], produces the opposite effects in OPC cultures, i.e., it increases I_K_ [50] and stimulates OPC maturation [50,51]. The fact that both A_2A_Rs and A_2B_Rs are Gs-coupled whereas GPR17 is a G_i_-coupled receptor, let us hypothesize that cAMP is involved in the myelination process. Indeed, when we applied the AC activator forskolin to patch-clamped cultured OPCs, we observed the same effect as A_2_R agonists, i.e., a decrease in I_K_. Furthermore, a subsequent BAY60-65683 application in the continuous presence of forskolin was devoid of effects [43], thus confirming that cAMP increase is responsible at least for A_2B_R-mediated inhibition of I_K_ and cell maturation. Results that forskolin inhibit I_K_ and cell maturation were reported in the nineties by Soliven and co-workers on cultured ovine OPCs [52].

Interestingly, an interplay occurs in cultured OPCs between A_2B_Rs and sphingosine kinase 1 (SphK1), one of the enzymes devoted to the synthesis of sphingosine 1 phosphate (S1P). This bioactive lipid mediator is reported to act as a mitogen in OPC cells [43]. We demonstrated that the A_2B_R agonist BAY60-6583 activates SphK1, thus rising S1P production, whereas its silencing by small interference RNA (siRNA) increases the expression of S1P lyase, the enzyme catalyzing irreversible S1P degradation inside the cells [43]. This observation led to hypothesize that the anti-differentiating effect exerted by A_2B_R activation in OPCs is mediated by an increase in S1P intracellular levels, as confirmed by findings that the SphK inhibitors VPC96047 or VPC96091 markedly increased MAG and MBP expression and also significantly reduced I_K_ currents in cultured OPCs [43].

Data about an inhibitory role of A_2B_R in myelin formation are consistent with recent findings from Manalo et al. [53] who demonstrated that elevated cochlear adenosine levels in ADA^−/−^ mice is associated with sensorineural hearing loss (SNHL) due to cochlear nerve fiber demyelination and mild hair cell loss. Intriguingly, A_2B_R-specific antagonists administered in ADA^−/−^ mice significantly restored auditory capacity, nerve fiber density, and myelin compaction [53]. The same authors also provided genetic evidence for A_2B_R upregulation not only in ADA^−/−^ hearing-impaired mice but also in age-related SNHL [53].

## 5. Conditions of Tissue Damage: Brain Ischemia

Ischemic stroke is the second leading cause of death in industrialized countries and the major cause of long-lasting disabilities worldwide [54]. Current treatments are confined to promote blood fluidity after the insult by the administration of tissue plasminogen activator (tPA) within the first phases (4–4.5 h) after stroke onset [55] and are efficient only in a restricted time window. The considerable socioeconomic impact of this pathology, together with the lack of effectiveness of current treatments, emphasizes the urgent need for new therapeutic targets able to prevent/repair brain damage.

Brain ischemia results from a permanent or transient reduction in cerebral blood flow mostly due to the occlusion of a brain artery. The consequent reduction of blood and/or oxygen supply to the brain leads to neuronal death caused by excessive glutamate release [56]. This early excitotoxic damage is followed by a secondary chronic phase of neuroinflammation that develops hours and days after ischemia. Either or both of these deleterious processes are important therapeutic targets.

Brain ischemia triggers a strong inflammatory response involving damage to the endothelium and to the blood–brain barrier (BBB), early recruitment of granulocytes, and delayed infiltration into the ischemic areas and the boundary zones by T cells and macrophages. This inflammatory *scenario* is mainly generated by necrotic cells, reactive oxygen species (ROS) generation, and numerous other factors caused by blood flow interruption in the brain. Once activated, these initiators of inflammation lead to numerous responses among which is the activation of microglia, the brain’s resident immune cells. Microglia then generate more proinflammatory cytokines [57] which in turn leads to adhesion molecule induction in the cerebral vasculature. These are the conditions under which adenosine is typically released in great amounts and may activate all subtypes of P1 adenosine receptors.

Protracted neuroinflammation is now recognized as the predominant mechanism of secondary ischemic damage [58]. Thus, besides the approved treatment with tPA in the first hours after ischemia, an important strategy to counteract the ischemic damage is to control brain injury progression after ischemia. Among neuromodulators involved in this event, adenosine has been recognized as a front line endogenous mediator for anti-inflammatory responses.

During ischemia, adenosine is released in massive amounts [35,59] and has long been known to act predominantly as an endogenous neuroprotectant agent [2,60]. Indeed, adenosine infusion into the ischemic striatum has been shown to significantly ameliorate neurological outcome and reduce infarct volume after transient focal cerebral ischemia [61]. This well recognized neuroprotection by adenosine during ischemia is principally ascribed to A_1_R stimulation [62], as they emerged as potent inhibitors of excitatory synaptic transmission both in vitro [63,64,65] or in vivo [66], as mentioned above. In particular, A_1_Rs counteract overstimulation of N-methyl D-aspartate (NMDA) receptors due to excessive glutamate release and consequent intracellular Ca^2+^ overload [11]. Among favorable consequences of A_1_R stimulation during ischemic insults are also reduction in cell metabolism and energy consumption [67], and a moderate hypothermia [68,69,70]. Unfortunately, the use of adenosine A_1_R agonists in ischemia is hampered by profound central and peripheral side effects such as bradycardia and sedation [70,71,72].

Of note, high levels of the enzyme AK, which recycles and removes adenosine by phosphorylation to form AMP, results in a decrease in the ambient adenosine and thus in reduced P1 receptor activation. The prompt ability of AK to alter adenosine availability has been recently described by Boison and Jarvis to provide a “site and event” specificity to the endogenous protective effects of adenosine in situations of cellular stress [73].

## 6. A_2B_Rs and Brain Ischemia

As mentioned above, A_2B_Rs are activated by µM concentrations of adenosine in tissues that experience ischemia, trauma, inflammation, or other types of stressful insults and, interestingly, mRNA and protein expression of A_2B_R increase to a greater extent after ischemia-reperfusion than does expression of the other three adenosine receptors [74]. Few works have investigated the role of A_2B_Rs in brain ischemia up to now because of the low potency of adenosine for the receptors and the few selective ligands developed so far.

Since it is known that A_2B_Rs enhance glutamate release in the CA1 hippocampus [37,38], one of the brain areas more susceptible to ischemic insults, the block of A_2B_Rs may be neuroprotective as it counteracts glutamate overload by preserving the inhibitory role of A_1_Rs on neurotransmission [37,38,40]. This is indeed the case in an in vitro model of brain ischemia reproduced in rat hippocampal slices by oxygen and glucose deprivation (OGD), as demonstrated by our group of research. We recently showed that the selective block of A_2B_Rs by the prototypical antagonist PSB-603 (50 nM) and by MRS1754 (200 nM) prevents irreversible synaptic failure produced by a severe, 7 min, OGD event in CA1 hippocampal slices. We also showed, in the same work, that anoxic depolarization (AD), an unequivocal sign of glutamate-induced excitotoxicity during OGD [75], is completely abolished in A_2B_R antagonist-treated slices exposed to 7 min OGD and is significantly delayed in slices undergoing a 30 min OGD insult. These results were accompanied by immunohistochemical analysis revealing that the A_2B_R block also counteracts the reduction of neuronal density found in CA1 stratum pyramidale at 3 h after OGD insults, decreases apoptosis, and maintains activated mTOR levels similar to those of controls, thus sparing neurons from the degenerative effects caused by the injury. Moreover, astrocytes significantly proliferate in CA1 stratum radiatum at 3 h after the end of OGD, possibly due to increased glutamate release [40]. A_2B_R antagonism significantly prevents astrocyte modifications. Of note, neither A_2B_R antagonist tested protects CA1 neurons from the neurodegeneration induced by exogenous glutamate application, indicating that the antagonistic effect is upstream of glutamate release, in line with data indicating a presynaptic effect of A_2B_Rs on glutamatergic terminals [37,38].

We consolidated the concept of A_2B_Rs acting through inhibition of presynaptic A_1_R subtype by showing that the application of a brief, reversible, synaptic episode of 2 min duration delays the reduction of evoked field potentials during OGD ascribed to A_1_R activation [38].

However, beyond neuroprotection exerted by A_2B_R antagonists acting at the neuro-glial level, it is worth noting that the bulk of evidence in the literature points to a beneficial role exerted by A_2B_R agonists acting on the same receptor subtype expressed also on peripheral blood cells [32,34]. Indeed, studies in mice ablated of A_2B_Rs on bone marrow cells indicate an important contribution of vascular A_2B_Rs in attenuating vascular leakage during hypoxia [34]. It was also found that activation of A_2B_Rs in a model of femoral artery injury is vasoprotective as it reduces myocardial infarct size in rabbit and mouse hearts when administered before or at the onset of reperfusion [32].

Of note, post-treatment with intravenous BAY60-6583 (1 mg/kg) reduces lesion volume in the absence or presence of tPA (10 mg/kg) and attenuates brain swelling, blood–brain barrier disruption, and tPA-exacerbated hemorrhagic transformation (HT) at 24 h after ischemia induced by transient (2 h) middle cerebral artery occlusion (tMCAo) [74]. Additionally, in the same work, BAY60-6583 mitigates sensorimotor deficits in the presence of tPA and inhibits tPA-enhanced matrix metalloprotease-9 activation, thus decreasing BBB permeability 24 h after ischemia [74]. Protection from BBB permeability after ischemia might protect from blood cell infiltration, that on their turn promote expansion of the inflammatory response in the ischemic tissue [25].

Our group of research contributed to the field by demonstrating that the chronic treatment with BAY60-6583, administered intraperitoneally twice/day for 7 days at the dose of 0.1 mg/kg, from 4 h after focal ischemia induced by tMCAo, since one day after ischemia protects from neurological deficit. Seven days after ischemia it protects from ischemic brain damage, neuronal death, microglia activation, and astrocyte alteration [76]. Interestingly, 7 days after ischemia, the A_2B_ agonist decreases TNF-α and increases IL-10 levels in the blood [76]. Both cytokines are considered valuable blood markers of the brain damage following an ischemic insult [77]. Among putative mechanism mediating protective effects of A_2B_ agonists, it is worth mentioning that A_2B_R agonists reduce the expression of TNF-α in primary microglia cultures [78] and increase IL-10 production from murine microglial cells [79] with consequent rescuing of the resting state of microglia. Besides a protection due to a direct agonism of A_2B_R located on rat microglial and/or astrocytic cells, observation that 2 days after tMCAo, BAY60-6583 significantly reduces granulocyte infiltration in the cortex [76] and supports the idea that A_2B_R activation on peripheral endothelial and blood cells is involved in counteracting inflammation of brain parenchyma. This possibility is also supported by the evidence that adenosine A_2B_R knock out (KO) mice show increased basal levels of TNF-α and expression of adhesion molecules in lymphoid cells, resulting in increased leukocyte rolling and adhesion [34]. Actually, increasing evidence indicates a role for A_2B_R in the modulation of inflammation and immune responses in distinct pathologies like cancer, diabetes, as well renal, lung, and vascular diseases [80]. Indeed, stroke and inflammation are strictly interrelated. Brain ischemia induces profound inflammatory changes in peripheral organs (especially lungs and gut) as early as 2 h after tMCAo in mice as detected by whole body single photon emission computed tomography (SPECT)-based imaging protocols [81]. Such peripheral inflammatory changes, in turn, may contribute to poorer recovery after stroke [81]. Data suggest that the A_2B_R agonists can be proposed as adjuvant therapy to the accepted pharmacological strategy with tPA in brain ischemia.

Taken together, data point toward the possibility that stimulation of A_2B_R plays a dual time-related role after ischemia. In the early hours after ischemia, a robust and sustained increase in cerebral extracellular levels of adenosine able to activate low affinity A_2B_Rs in the brain may contribute to expand excitotoxicity. However, in the hours and days following ischemia, when profound neuroinflammation develops, A_2B_Rs located on glial, vascular endothelial, and blood cells exert a prevalent immunomodulatory role attenuating the neuroinflammation. Thus, on the whole, it appears that A_2B_Rs located on any cell type of the brain and on vascular and blood cells partake in either salvage or demise of the tissue after a stroke and represent an important target for drugs that have different therapeutic time-windows after stroke.

Since it is known that an A_2B_R agonist administered within the first 4 h after brain ischemia is able to reduce neurological deficit measures at 24 h after the insult [76], we could hypothesize that a beneficial effect of A_2B_R activation could be achieved by administration of a selective agonist starting at 1 day after stroke. Table 1 summarizes A_2B_R-mediated effects in in vitro or in vivo experimental models of brain ischemia.

In recent years, a similar dual role of the other subtype of A_2_Rs, the A_2A_R, was described. Indeed, antagonists at this receptor proved neuroprotective when applied during in vitro OGD [84,85] or in the acute post-ischemic phase after MCAo in rats [86,87,88,89], whereas the same receptor agonists proved protective at later phases after ischemia by decreasing neuroinflammation [35,90]. This “paradoxical” role of the A_2A_Rs has been discussed in several review papers [35,91].

Hence, it appears that the reciprocal influence of central (neuro-glia) versus peripheral (blood vessels–inflammatory cells) mechanisms involved in brain ischemia are hard to dissect.

What recently and interestingly emerged about the A_2B_R subtype is that it might be a “sensor” of blood oxygen levels and rescue hypoxic tissues by an additive mechanism involving the release of oxygen from erythrocytes. Indeed, recent findings obtained in humans challenged with altitude indicate that A_2B_Rs expressed on erythrocytes elevate their O_2_ releasing capacity by increasing 2,3-biphosphoglycerate (2,3-BPG) levels in an AMP-activated protein kinase-dependent manner [92]. Of note, the same authors also reported a significant increase in plasma adenosine concentrations and soluble CD73 activity in 21 healthy humans within 2 h of arrival at 5260 m altitude [92]. These findings unveiled a novel mechanism of human adaptation to hypoxia, in this case due to high altitude, and pointed to A_2B_Rs as possible targets to facilitate O_2_ release capacity to peripheral tissues and a potential therapeutic approach for counteracting hypoxia-induced tissue damage.

## 7. A_2B_Rs and Demyelinating Diseases

Demyelination occurs in a variety of pathological conditions affecting central or peripheral nervous systems. As an example, myelin disorganization in caudate/putamen striatal nuclei have been reported by us [88] and others [93,94]. Furthermore, chronic demyelinating diseases, such as multiple sclerosis (MS), are highly invalidating pathologies with elevated incidence among the “under 40” population worldwide [95], but an efficacious therapy is still lacking.

It is interesting to note that elevated adenosine levels have been detected in cerebrospinal fluid of MS patients [96,97]. Furthermore, an upregulation of A_2A_R [98] and A_2B_R [83] has been reported in peripheral blood leukocytes of MS patients and in the CNS [82] and peripheral lymphoid tissues [83,99] in a mouse model of MS, the experimental autoimmune encephalomyelitis (EAE).

Hence, a crucial role of adenosine, and in particular of A_2A_R and/or A_2B_R subtypes, in demyelinating pathologies have been postulated. Under these conditions, excessive signaling by excitatory neurotransmitters like glutamate may be deleterious to neurons and oligodendroglia by exacerbating excitotoxicity and contributing to brain injury. For this reason, the inhibitory effect on glutamate release described above for antagonists at both A_2_R subtypes could prove protective. This was indeed the case, as demonstrated by Chen and colleagues [100] and by Wei and co-workers [83] who reported that A_2A_R and A_2B_R antagonists, respectively, alleviated the clinical symptoms of EAE and prevented demyelination and CNS damage. Recent data by Liu and co-workers [82] confirmed that A_2B_R activation seems to participate in EAE-induced damage as BAY60-6583 reverted the protective effects, i.e., reduced inflammatory cell infiltration and demyelination, exerted by mesenchymal stem cell therapy in EAE mice. Of note, the above results demonstrating a deleterious role of A_2B_Rs in demyelinating diseases are in agreement with our in vitro data demonstrating that A_2B_R blockade [44], as well as A_2A_R antagonism [45], facilitates OPC differentiation in vitro.

However, things are probably more complicated as suggested by the fact that, again, A_2_R-mediated actions are mainly anti-inflammatory when observed in a longer time-span. Indeed, genetically modified A_2A_R^-/-^ EAE mice are more prone to EAE-induced damage [45], and the A_2A_R agonist CGS61680 ameliorates EAE by reducing Th1 lymphocyte activation and cytokine-induced BBB dysfunction [101]. Indeed, adenosine receptors are expressed also by infiltrating lymphocytes, macrophages, and microglial cells and, accordingly, many works suggest that the possible beneficial effects linked to this pathway are mainly related to the immune-modulating consequences rather than to a remyelinating or neuroprotecting effect. The dual role of A_2A_Rs in demyelinating diseases has been elegantly commented by Rajasundaram who tried to reconcile the apparently paradoxical data reported up to now about A_2A_Rs in demyelinating diseases [102].

Unfortunately, data about the effects of A_2B_R ligands in demyelinating pathologies are not that abundant yet, as shown in Table 1.

## 8. Conclusions

In conclusion, results underlie that after hypoxia/ischemia, brain injury results from a complex sequence of pathophysiological events that evolve over time—a primary acute mechanism of excitotoxicity and periinfarct depolarizations followed by a secondary brain injury activation triggered by protracted neuroinflammation. Information so far acquired indicates that adenosine A_2B_Rs located on any cell type of the brain and on vascular and blood cells partake in either salvage or demise of the tissue after a stroke, including protracted demyelination.

Thus, they all represent important targets for drugs having different therapeutic time-windows after stroke. The final protective outcome for an agonist versus antagonist compound depends on time of administration and district of activation of the receptor targeted by the drug.

## Figures and Tables

**Table 1 ijms-21-09697-t001:** Effects of adenosine A_2B_R receptor (A_2B_R) activation or blockade in different in vitro or in vivo experimental approaches.

A_2B_R-Mediated Effects	In Vitro Ischemia in Rat Hippocampal Slices (OGD)	In Vivo Ischemia Model (MCAo) in the Rat	In Vivo Model of Multiple Sclerosis (EAE) in the Mouse
**A_2B_R activation**	↓ A_1_R-mediated inhibition of Glu release [38]	Protects from brain damage, neurological deficit [76], and BBB disruption [74]	Reverses MSC-mediated BBB repair [82]
**A_2B_R block**	Protects from AD and synaptic failure [40]	unknown	Protects from myelin loss and neurological damage [83]

Glutamate: Glu. Anoxic depolarization: AD. Oxygen and glucose deprivation: OGD. Middle cerebral artery occlusion: MCAo. Experimental autoimmune encephalomyelitis: EAE. Mesenchymal stem cells: MSCs.

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
