# Peer review of "A_2B_ Adenosine Receptors: When Outsiders May Become an Attractive Target to Treat Brain Ischemia or Demyelination"

_ijms, 2020, doi:10.3390/ijms21249697_

Round 1
Reviewer 1 Report
The MS by Coppi et al. is a timely review of the function of a subtype of adenosine receptor, namely the one termed A2B, in physiological and pathological conditions. While such receptor has been long overlooked in the field, its role is becoming increasingly important. This is why the present review should be of wide interest.
The MS is generally well-written with its focus on the mechanistic explanation of receptor mediated effects, and relies on the strong contribution by the authors to this research area.
There are a few minor issues that the authors are advised to deal with in order to improve the strength of their contribution.
- The title is somewhat misleading because strategies for brain ischemia are certainly not attractive. Perhaps something like “may become an attractive target to treat brain ischemia or demyelination”
Lines 17 and 21 “role in” instead of “role on”.
The MS would greatly benefit from adding a Figure with a time-related scheme for the various contributions of A2B to the progression of ischemia-induced effects.
Lines 186-7 Sentence starting with “protracted inflammation…” needs a reference.
Lines 200-1 what side effects? Please provide a reference.
In view of the multifarious action of A2B on early and late ischemic events, it is important to discuss how to decide when to administer one AD2B agonist after the initial ischemic event. This point should be better explained on p. 7 (1st para) ideally with reference to a Figure as proposed above. If the authors have any hypothesis on this important issue which is the basis of their review, it would be helpful to provide it here.
A list of abbreviations for enzymes and receptors mentioned in the review would be useful.
Author Response
Reviewer 1
The MS by Coppi et al. is a timely review of the function of a subtype of adenosine receptor, namely the one termed A2B, in physiological and pathological conditions. While such receptor has been long overlooked in the field, its role is becoming increasingly important. This is why the present review should be of wide interest.
The MS is generally well-written with its focus on the mechanistic explanation of receptor mediated effects, and relies on the strong contribution by the authors to this research area.
Our response. We thank the reviewer for his/her positive comments and suggestion that significantly improved our work.
There are a few minor issues that the authors are advised to deal with in order to improve the strength of their contribution.
- The title is somewhat misleading because strategies for brain ischemia are certainly not attractive. Perhaps something like “may become an attractive target to treat brain ischemia or demyelination”
Our response. We modified the title as suggested.
Lines 17 and 21 “role in” instead of “role on”.
Our response. We modified the sentences as suggested.
The MS would greatly benefit from adding a Figure with a time-related scheme for the various contributions of A2B to the progression of ischemia-induced effects.
Lines 186-7 Sentence starting with “protracted inflammation…” needs a reference.
Our response. The following reference was added, as suggested (page 6 line 193):
“Wang X, Xuan W, Zhu ZY, Li Y, Zhu H, Zhu L, Fu DY, Yang LQ, Li PY, Yu WF. The evolving role of neuro-immune interaction in brain repair after cerebral ischemic stroke. CNS Neurosci Ther. 2018 Dec;24(12):1100-1114. doi: 10.1111/cns.13077. Epub 2018 Oct 22.”
Lines 200-1 what side effects? Please provide a reference.
Our response. The following sentence and references were added, as suggested (page 7 line 208):
“…. by profound central and peripheral side effects such as bradycardia and sedation (M I Sweeney. Neuroprotective effects of adenosine in cerebral ischemia: window of opportunity. Neurosci Biobehav Rev 1997 Mar;21(2):207-17; D K Von Lubitz 1, R C Lin, N Melman, X D Ji, M F Carter, K A Jacobson. Chronic administration of selective adenosine A1 receptor agonist or antagonist in cerebral ischemia. Eur J Pharmacol 1994 Apr 21;256(2):161-7; A Sollevi. Cardiovascular effects of adenosine in man; possible clinical implications. Prog Neurobiol 1986;27(4):319-49)
In view of the multifarious action of A2B on early and late ischemic events, it is important to discuss how to decide when to administer one AD2B agonist after the initial ischemic event. This point should be better explained on p. 7 (1st para) ideally with reference to a Figure as proposed above. If the authors have any hypothesis on this important issue which is the basis of their review, it would be helpful to provide it here.
Our response. Due to multiple actions of A2BRs, we agree with the reviewer that an additional Figure would improve readability of the present review. However, in spite of abundant data in the literature about A2ARs ligands given at different time after ischemia, a paucity of works is available concerning the A2BR subtype. Indeed, only two works are present that describe A2BR effects in MCAo models, and in both of them A2BR agonists are administered. For this reason, to avoid overestimation of available data, we added a new table (see Table 1), rather than a figure. We also added the following sentences at page 9 lines 294-298:
“Since it is known that an A2BR agonist administered within the firsts 4 h after brain ischemia is able to reduce neurological deficit measure at 24h after the insult, we could hypothesize that a beneficial effect of A2BR activation could be achieved by administration of a selective agonist starting at 1 day after stroke. Table 1 summarizes A2BR-mediated effects in in vitro or in vivo experimental models of brain ischemia.”
A list of abbreviations for enzymes and receptors mentioned in the review would be useful.
Our response. A list of abbreviations has been added.

Reviewer 2 Report
The authors have undertaken a fairly comprehensive review of A2B adenosine receptors involved in oligodendrogliogenesis, brain ischemia, and demyelination. The authors concluded that A2B adenosine receptors may be therapeutic targets for the still unmet treatment of stroke or demyelinating diseases. I have some comments that I believe need to be addressed prior to publication of this article.
Minor comments:
Page 4 lines 132–134, “We also demonstrated that A2BR activation reversibly inhibits tetraethylammonium- (TEA-) sensitive, sustained IK, and 4-amynopyridine- (4-AP) sensitive, transient IA, conductances [42].”, I guess [43].
References 36 and 57 are not cited in the manuscript.
Author Response
Reviewer 2
The authors have undertaken a fairly comprehensive review of A2B adenosine receptors involved in oligodendrogliogenesis, brain ischemia, and demyelination. The authors concluded that A2B adenosine receptors may be therapeutic targets for the still unmet treatment of stroke or demyelinating diseases. I have some comments that I believe need to be addressed prior to publication of this article.
Our response. We thank the reviewer for his/her positive comments and suggestion that significantly improved our work.
Minor comments:
Page 4 lines 132–134, “We also demonstrated that A2BR activation reversibly inhibits tetraethylammonium- (TEA-) sensitive, sustained IK, and 4-amynopyridine- (4-AP) sensitive, transient IA, conductances [42].”, I guess [43].
Our response. We thank the reviewer for let us noticing the mistake. The correct reference (Coppi et al., Biochem Pharmacol 2020) has been added (page 5 line 141).
References 36 and 57 are not cited in the manuscript.
Our response. We apologies for the mistake. These references have been deleted.
